# Faster Short-Chain Fatty Acid Absorption from the Cecum Following Polydextrose Ingestion Increases the Salivary Immunoglobulin A Flow Rate in Rats

**DOI:** 10.3390/nu12061745

**Published:** 2020-06-11

**Authors:** Yuko Yamamoto, Toshiya Morozumi, Toru Takahashi, Juri Saruta, Masahiro To, Wakako Sakaguchi, Tomoko Shimizu, Nobuhisa Kubota, Keiichi Tsukinoki

**Affiliations:** 1Department of Junior College, School of Dental Hygiene, Kanagawa Dental University, 82 Inaoka, Yokosuka 2388580, Kanagawa, Japan; yamamoto.yuko@kdu.ac.jp; 2Division of Periodontology, Department of Oral Interdisciplinary Medicine, Graduate School of Dentistry, Kanagawa Dental University, 82 Inaoka, Yokosuka 2388580, Kanagawa, Japan; morozumi@kdu.ac.jp; 3Department of Health and Nutrition, Faculty of Human Health, Kanazawa Gakuin University, 10 Sue-machi, Kanazawa 9201392, Ishikawa, Japan; t-takahasi@kanazawa-gu.ac.jp; 4Division of Environmental Pathology, Department of Oral Science, Graduate School of Dentistry, Kanagawa Dental University, 82 Inaoka, Yokosuka 2388580, Kanagawa, Japan; saruta@kdu.ac.jp (J.S.); sakaguchi@kdu.ac.jp (W.S.); n.kubota@kdu.ac.jp (N.K.); 5Division of Dental Anatomy, Department of Oral Science, Graduate School of Dentistry, Kanagawa Dental University, 82 Inaoka, Yokosuka 2388580, Kanagawa, Japan; m.tou@kdu.ac.jp; 6Department of Highly Advanced Stomatology, Graduate School of Dentistry, Kanagawa Dental University, 3-31-6 Tsuruya, Kanagawa-ku, Yokohama 2210835, Kanagawa, Japan; shimizu@kdu.ac.jp

**Keywords:** IgA, saliva, short-chain fatty acids, polydextrose, polymeric immunoglobulin receptor, rats

## Abstract

Salivary immunoglobulin A (IgA) plays a vital role in preventing upper respiratory tract infections (URTI). In our previous study, we showed that the intake of carbohydrates increases the intestinal levels of short-chain fatty acids (SCFAs), which in turn increase salivary IgA levels. However, the mechanism underlying this phenomenon has not been fully elucidated. In this study, we investigated in rats the effect of polydextrose (PDX) ingestion on salivary IgA level and SCFA concentration in cecal digesta and the portal vein. Five-week-old rats were fed with a fiber-free diet (control) or with 40 g/kg of PDX for 28 days. Compared to the control, ingestion of PDX led to a higher salivary IgA flow rate (*p* = 0.0013) and a higher concentration of SCFAs in the portal vein (*p* = 0.004). These two data were positively correlated (r_s_ = 0.88, *p* = 0.0002, n = 12). In contrast, the concentration of SCFAs in cecal digesta and cecal digesta viscosity were significantly lower following PDX ingestion, compared to the control (*p* = 0.008 and 0.05, respectively). These findings suggest that the ingestion of PDX increases the absorption rate of SCFAs in the intestine through PDX-induced fermentation, which is accompanied by an increase in SCFA levels in the blood, and ultimately leads to increased salivary IgA levels.

## 1. Introduction

The oral cavity is the entry point to the respiratory and gastrointestinal systems [1]. The oral cavity is always in contact with the outside environment and is more at risk of infection by exogenous microorganisms [2]. To prevent infections, the oral cavity is covered by the mucosal epithelium and is protected by the mucosal immune system [3]. In addition, the mucous membranes of the oral cavity and pharynx are protected by a constant saliva flow. Saliva is produced from three major salivary glands and hundreds of minor salivary glands located in the oral cavity [4].

Of the many functions of the saliva (lubrication, buffer), the antibacterial function is one of the most important [4]. Although 99% of the saliva is composed of water, the remaining organic components include many antibacterial substances such as lysozyme, peroxidase, lactoferrin, β-defensin, and immunoglobulin A (IgA) [5]. Salivary IgA plays an important role in preventing infection, as it specifically binds to pathogenic bacteria and viruses to prevent them from entering the oral mucosa and neutralizes toxins [6]. Dimeric IgA, produced by plasma cells in the salivary gland, is transported to the lumen by the polymeric immunoglobulin receptor (pIgR), expressed on the basolateral surface of epithelial cells [7]. Therefore, pIgR is an important mediator for the secretion of IgA into saliva [8]. It has been reported that reduced levels of IgA in the saliva increase the risk of upper respiratory tract infections (URTI), and vice versa [9,10]. Elderly people, children, and patients with diabetes who have poor immune function are at increased risk of URTI infection [11,12,13]. Therefore, increasing pIgR expression and salivary IgA levels in patients with low immunity are important for URTI prevention.

Mucosal immunity is activated by the ingestion of indigestible carbohydrates [14]. Many studies reported that the ingestion of indigestible carbohydrates reshapes the ecology of intestinal bacteria and increases short-chain fatty acids (SCFAs) and IgA production in the large intestine, activating mucosal immunity [14,15,16]. Ingestion of fructooligosaccharides (FOS), a class of indigestible carbohydrates, increases the expression of both IgA and pIgR in the rat cecum [17]. Moreover, it has been reported that, following FOS digestion, the expression of cytokines such as interferon γ (IFN-γ) and interleukin 17A (IL-17A) is increased [18]. Polydextrose (PDX), like FOS, is a type of indigestible carbohydrate that stimulates IgA production in the large intestine [19]. Furthermore, it has been reported that the ingestion of PDX increases the IgA concentration in cecal digesta but simultaneously decreases the SCFA concentrations [20]. These results were replicated in our previous study, in which rats were fed a diet containing a combination of PDX and lactitol [21].

The effect of the intake of indigestible carbohydrates on intestinal IgA levels is known and well-studied; however, the effects on salivary IgA production are poorly understood. In our previous study, we showed that FOS intake increased the SCFA concentration in cecal digesta, the salivary IgA flow rate, and the pIgR expression levels in the submandibular gland. We hypothesized that the SCFA increase in the rat cecum following FOS intake is likely to be involved in the increase of salivary IgA flow rate [22,23]. In contrast, rats fed a combination of PDX and lactitol showed an increased salivary IgA flow rate but also decreased SCFA levels in cecal digesta [21]. Furthermore, the effect of PDX ingestion on the pIgR expression levels remains unknown. It has been elucidated that increased concentration of SCFAs in cecal digesta due to indigestible carbohydrates intake may increase the rate of salivary IgA secretion. However, it is unclear whether the concentration of SCFAs in cecal digesta or the SCFAs absorption rate from the cecum is responsible for the regulation of salivary IgA expression and flow rate. In addition, the mechanism by which the ingestion of indigestible carbohydrates increases pIgR in the salivary glands is not clear. Therefore, in the present study, we investigated whether the ingestion of PDX alone can increase the IgA flow rate in the saliva and the pIgR expression level in the salivary gland. Moreover, we investigated whether the increase in the salivary IgA flow rate was due to the concentration of SCFAs in cecal digesta or the SCFA absorption from the cecum.

## 2. Materials and Methods

### 2.1. Animals

A total of 12 male Wistar rats (4 weeks old) were obtained from CLEA Japan (Tokyo, Japan) and housed individually in wire mesh cages without bedding material at 22 ± 3 °C in a 12 h light–12 h dark cycle. The rats had free access to the fiber-free control diet (see Appendix A) and water for 7 days before the start of the experiment. Thereafter, we randomly divided the 12 rats into two groups (Control and PDX). The groups were maintained under equivalent conditions except for diet composition. During the experiment, rats had free access to diet and water. The feed intake of rats was determined by measuring the weight of the residual feed every three days.

The protocol for this study was reviewed and approved by the Ethics Committee for Animal Experiments of Kanagawa Dental University (approval number: 17-001, approval date: April 4 2016) and was performed following the Guidelines for Animal Experimentation of Kanagawa Dental University and the Animal Research: Reporting of In Vivo Experiments (ARRIVE) guidelines for reporting animal research.

### 2.2. Diets

The compositions of the fiber-free control diet and the experimental diet (PDX) are shown in Appendix A. The nutrient composition of both diets and the rationale for their use were the same as adopted in our previous study [23]. PDX (Litesse^®^ UltraTM; Danisco Japan Ltd., Tokyo, Japan) was added at a concentration of 40 g/kg to the control diet, which was prepared by Japan CLEA. In a previous study by Sepehr et al., rats were fed 50 g/kg of PDX [24]. However, we used a lower concentration of 40 g/kg of PDX because, in our previous study, rats suffered from diarrhea for several days after being fed 25 g/kg of PDX and 25 g/kg of lactitol [21].

### 2.3. Sampling

After 28 days, all rats were anesthetized with isoflurane (3–4% for the induction, 1–2% for the maintenance; Mylan Inc., Tokyo, Japan). Under anesthesia, saliva and portal vein blood samples were collected. Portal vein blood samples were collected by cardiac puncture into Venoject II tubes containing Serum Separating Medium (Terumo Corporation, Tokyo, Japan). The tubes were immediately inverted five to six times and placed at 25 °C for 3 h, after which they were centrifuged (1200× *g*, 20 min, 20 °C). After sampling, rats were euthanized by decapitation and the cecal digesta, cecal tissues, and submandibular glands were excised. All samples were collected no later than 4 h after the anesthesia, weighed immediately, and stored at −80 °C until further analysis.

### 2.4. Collection of Saliva

Saliva of the rats was collected using the same protocol described in our previous study [22]. The whole saliva flowed out by pilocarpine (8 mg/kg body weight) injection was collected with a micropipette for 10 min.

### 2.5. Measurement of IgA Concentration

The concentration of IgA in the saliva was measured using the same protocol described in our previous study [23]. Saliva weight (g) collected for 10 min was measured, and the specific gravity of saliva was assumed to be 1.00 g/mL, and the salivary flow rate (mL/10 min) was calculated. Salivary IgA flow rate (μg/10 min) was calculated by multiplying salivary IgA concentration (μg/mL) by salivary flow rate (mL/10 min) [23]. The salivary IgA flow rate normalized on the submandibular gland tissue weight (µg/10 min/g) was calculated as described in our previous studies [21,23].

### 2.6. Measurement of IFN-γ Concentration

The concentration of IFN-γ in the submandibular gland tissue was quantified by using the Rat IFN-γ ELISA kit pink-ONE kit (Koma Biotech Inc., Seoul, South Korea). A 96-well microtiter plate pre-coated with antigen affinity-purified mouse anti-rat IFN-γ antibody was washed four times with a wash solution (8 Na2HPO4, 150 NaCl, 2 KH2PO4, 3 mM KCl, and 0.5 g/L Tween^®^ 20). The same solution was used for all the subsequent washes. Samples and rat IFN-γ standards were added to the plate. Then, the plate was incubated for 2 h at 25 °C and washed four times. A biotinylated antigen affinity-purified mouse anti-rat IFN-γ antibody was added to the plate incubated for 2 h at 25 °C. Afterward, the plate was washed four times. Horseradish peroxidase (HRP)-conjugated streptavidin was added to the plate and incubated for 30 min at room temperature. After four washes, the solution containing the enzyme substrate 3,3′,5,5′-tetramethylbenzidine (TMB) was added to the plate. The plate was then incubated in the dark at 25 °C for 10 min, after which the reaction was stopped with a 2M H2SO4 solution. The 450 nm absorbance was measured by an automated microplate reader (BioRad, Hercules, California, USA).

### 2.7. Measurement of IL-17A Concentration

The concentration of interleukin IL-17A in the submandibular gland tissue was quantified using the Rat IL-17A Platinum ELISA kit (Affymetrix, Inc., Santa Clara, CA, USA). A 96-well microtiter plate pre-coated with a mouse anti-rat IL-17A antibody was washed two times with a wash solution (8 Na_2_HPO_4_, 150 NaCl, 2 KH_2_PO_4_, 3 mM KCl, and 10 g/L Tween^®^ 20). The same solution was used for all the subsequent washes. Samples and rat IL-17A standards were added to the plate. Afterward, a biotin-conjugated anti-rat IL-17A monoclonal antibody was added to the plate and incubated for 2 h at 25 °C, after which the plate was washed four times. HRP-conjugated streptavidin was added to the plate and incubated for 1 h at 25 °C. After four washes, a solution containing the 3,3′,5,5′-TMB substrate was added to the plate. Then, the plate was developed in the dark at 25 °C for 30 min and the reaction was stopped with 1M of H_2_SO_4_. The 450 nm absorbance was measured by an automated microplate reader (BioRad, Hercules, CA, USA).

### 2.8. Measurement of Cecal Digesta Water Content

Before freezing, fresh cecal digesta were placed in 1.5 mL plastic tubes and dried in a constant temperature oven (Yamato Scientific Co., Ltd., Tokyo, Japan) at 135 °C for 3 h. Thereafter, the cecal samples were dried in a desiccator until a constant weight was reached. The weight of the cecal digesta was calculating the difference in weight before and after the drying process.

### 2.9. Measurement of Cecal Digesta Viscosity

Cecal digesta samples were taken within 10 min from the euthanization. The viscosity of fresh cecal digesta containing solid particles was measured using the digital cone-plate viscometer HBDV-Prime (AMETEK Brookfield, MA, USA) equipped with CPE51 and CPE52 spindles (AMETEK Brookfield, MA, USA), using shear rates between 0.3 and 100 S^−1^. The presence of solid particles is the main factor affecting the cecal digesta viscosity [25]. In the rat cecal digesta, the measurement of viscosity at a shear rate of 1 S^−1^ was taken as the physiological viscosity, as previously reported [26].

### 2.10. RNA Extraction, cDNA Synthesis, and Quantitative Real-Time PCR Measurement of pIgR mRNA in Submandibular Glands

The isolation of total RNA from the rat submandibular glands and cDNA synthesis was performed as previously reported [22]. Quantitative real-time PCR was performed using a LightCycler 480 system (Roche Diagnostics Limited, West Sussex, UK), as previously reported [22]. The primer sequences used to amplify the pIgR gene sequence were the following: 5′-TGG GAG CTA CAA GTG TGG TC-3′ (forward primer) and 5′-GGG TGT CAT TTG GGA ATC CAG-3′ (reverse primer). The TaqMan probe was designed and synthesized by Nippon Gene Research Laboratory (Miyagi, Japan) and has the following sequence: FAM-5′-TTC GAT GTC AGC CTG GAG GTC AGC-3′-TAMRA. As a control, the β-actin housekeeping gene was amplified using the LightCycler, FastStart DNA Master SYBR Green I kit (Roche Diagnostics Limited) following the manufacturer’s instructions (Nippon Gene Research Labs, Inc.). The primer pair used was the following: 5′-CTT GTA TGC CTC TGG TCG TA-3′ (forward primer), and 5′-CCA TCT CTT GCT CGA AGT CT-3′ (reverse primer). The primers were validated by performing a melting temperature analysis and by inspection of the DNA bands by agarose gel electrophoresis.

### 2.11. Measurement of SCFA in Cecal Digesta

SCFAs in cecal digesta were measured by high-performance liquid chromatography, as previously reported [23].

### 2.12. Portal Vein Blood Preparation

The preprocessing of portal vein blood samples for gas chromatography coupled with mass spectroscopy (GC-MS) analysis was adapted from a protocol reported by Tsukahara et al. [27].

### 2.13. Measurement of SCFA in Portal Vein Blood

Measurement of SCFA concentrations in portal vein blood was based on the method reported by Tsukahara et al. [27].

### 2.14. Bayesian Network

Bayesian network is a graphical model that describes the causal relationships between certain events based on their probability. The details on the use of this model to analyze the causal relationship between risk factors have been previously reported by Maglogiannis et al. [28].

### 2.15. Statistical Analysis

Statistical analyses were performed with JMP version 12 (SAS Institute Japan, Tokyo, Japan) and R version 3.2.0 (The R Project for Statistical Computing, Vienna, Austria, 2013). Results are presented as the mean plus the standard error of the mean (SEM). Comparisons between the two groups were analyzed using Welch’s *t*-test. Spearman’s rank correlation was employed to analyze the statistical significance of the correlation between two variables. The statistical analysis of cecal digesta viscosity was performed using the factorial two-way analysis of variance (ANOVA). The Tukey–Kramer method for interaction was used for post hoc analysis when the interaction calculated with ANOVA was significant. Causal effects between variables were evaluated using the Bayesian network analysis. *p* < 0.05 denoted statistical significance.

## 3. Results

### 3.1. Effect of PDX Ingestion on Salivary IgA Production and SCFA-Induced Changes

We observed no difference in the intake of foods between the control and PDX groups. Both groups ingested an average of 20 g of feed per day. Likewise, we observed no differences between the two groups in body weight gain (before the experiment, rats had similar body weights), mean left and right submandibular glands’ weight, and the 10 min salivary flow rate after the 28 day feeding period (*p* = 0.2, 0.8, and 0.7, respectively, Welch’s *t*-test, Appendix A).The cecum tissue weight, cecal digesta weight, and cecal digesta water content in the PDX group were higher than those in the control group (*p* < 0.0001, *p* < 0.0001, and *p* = 0.007, respectively, Welch’s *t*-test, Figure 1).

Salivary IgA concentration and flow rate normalized on the submandibular gland tissue’s weight in the PDX group were higher than those in the control group (*p* = 0.001 and 0.001, respectively, Welch’s *t*-test, Figure 2).

In the cecal digesta, the concentration of acetate and propionate, and total SCFAs in the PDX group were lower than those in the control group (*p* = 0.004, 0.001, and 0.008, respectively, Welch’s *t*-test, Table 1). In the portal vein blood samples, the concentration of acetate and n-butyrate, and total SCFAs in the PDX group were higher than those in the control group (*p* = 0.005, 0.03, and 0.004, respectively, Welch’s *t*-test, Table 2). The amount of acetate, n-butyrate, and total SCFAs in the cecal digesta in the PDX group were higher than those in the control group (*p* = 0.005, 0.003, and 0.007, respectively, Welch’s *t*-test, Table 3). We observed a significant interaction between PDX addition and cecal digesta viscosity shear rate (*p* < 0.001, two-way ANOVA, Figure 3). The viscosity in the PDX group was lower than that in the control group at 0.5, 1, and 2 S^−1^ shear rates (*p* < 0.05, Tukey–Kramer method post-hoc analysis, Figure 3).

The pIgR mRNA expression level in the submandibular gland tissue in the PDX group was higher than that of the control group (*p* = 0.02, Welch’s *t*-test, Figure 4A). Instead, we observed no differences between the two groups in the concentration of IFN-γ and IL-17A (*p* = 0.8 and 0.3, respectively, Welch’s *t*-test, Figure 4B,C).

### 3.2. Correlation between SCFAs, IgA, and pIgR Parameters

We found that the salivary IgA flow rate normalized by submandibular gland tissue weight positively correlated with SCFA concentration in the portal vein blood samples (r_s_ = 0.88, *p* = 0.0002, n = 12), the cecal tissue weight (r_s_ = 0.76, *p* = 0.004, n = 12), the cecal digesta water content (r_s_ = 0.76, *p* = 0.005, n = 12), pIgR expression level in the submandibular gland (r_s_ = 0.66, *p* = 0.02, n = 12), and cecal digesta weight (r_s_ = 0.60, *p* = 0.04, n = 12) (Table 4). In contrast, the salivary IgA flow rate was not correlated with the SCFA concentration in cecal digesta (r_s_ = -0.53, *p* = 0.09, n = 12, Table 4).

We observed that the SCFA concentration in portal vein blood samples positively correlated with the cecal tissue weight (r_s_ = 0.74, *p* = 0.006, n = 12), cecal digesta weight (r_s_ = 0.69, *p* = 0.01, n = 12), the amount of SCFAs in cecal digesta (r_s_ = 0.66, *p* = 0.02, n = 12), and cecal digesta water content (r_s_ = 0.63, *p* = 0.03, n = 12) (Table 5).

The pIgR expression level in the submandibular gland tissue positively correlated with the cecal tissue weight (r_s_ = 0.87, *p* = 0.0003, n = 12), cecal digesta weight (r_s_ = 0.77, *p* = 0.003, n = 12), and SCFA concentration in portal vein blood samples (r_s_ = 0.71, *p* = 0.009, n = 12) (Table 6). In contrast, the concentration of IFN-γ and IL-17A in the submandibular gland tissue were not correlated with the pIgR expression level (r_s_ = −0.046, −0.028 and *p* = 0.9, 0.9, respectively, n = 12, Table 6). Collectively, our results suggested that, in rats, PDX ingestion increased the salivary IgA flow rate, the pIgR expression level in the submandibular gland tissue, and the SCFA concentration in portal vein blood samples (Figure 2B and Figure 4A, Table 2). Furthermore, PDX ingestion increased the cecal digesta water content and decreased cecal digesta viscosity (Figure 1C and Figure 3).

### 3.3. Bayesian Network of the Causal Effects Induced by PDX Ingestion

To investigate the relationship between the different parameters measured in this study, we performed a Bayesian network analysis. The network showed that the salivary IgA flow rate normalized on the submandibular gland tissue weight was directly affected by the SCFA concentration in the portal vein (Figure 5). In agreement with the Bayesian network, we found that, upon PDX ingestion, the salivary IgA flow rate was highly correlated to the SCFA concentration in the portal vein (Table 4).

### 3.4. Factors Related to the Increase of SCFA Concentration in Portal Blood

The Bayesian network analysis also showed that SCFA concentration in the portal vein was directly affected by the cecal tissue weight and the amount of SCFAs in the cecal digesta (Figure 5). Consistent with this, we found higher levels of SCFAs in the portal vein in the PDX group compared to those in the control group (Table 2). Furthermore, the cecum tissue weight, cecal digesta weight, amount of SCFAs in cecal digesta, and cecal digesta water content were higher, while cecal digesta viscosity was lower in the PDX group than those in the control group (Figure 1 and Figure 3, Table 2). Similarly, we found a positive correlation between SCFA concentration in the portal vein and cecum tissue weight, cecal digesta weight, the amount of SCFAs in cecal digesta, and cecal digesta water content (Table 5). The consistency index and the flow behavior index of the cecal digesta represent the viscosity of the cecal digesta: The Bayesian network analysis showed that these indexes were directly affected by the cecal digesta water content (Figure 5).

## 4. Discussion

Our results show that the SCFA concentration in the portal vein reflects its absorption from the cecum into the bloodstream [29]. A higher cecal tissue weight indicates a higher cecal tissue surface area [30,31]. Conversely, it has been reported that an increase in the cecal surface area increased the cecum’s capacity for absorbing molecules [32]. At the same time, the ingestion of indigestible carbohydrates has been reported to increase the digesta residue water content in the large intestine [33]. Finally, it has been reported that when the cecal digesta water content decreases, the digesta viscosity increases, and vice versa [33,34]. This previous knowledge is consistent with our Bayesian network analysis, showing that the consistency and flow indexes, representing the cecal digesta viscosity, were directly affected by the cecal digesta water content (Figure 5). Furthermore, it has been reported that an increase of the digesta viscosity delays glucose diffusion in the lumen, decreases glucose absorption from the lumen, and decreases glucose concentration in the plasma [35]. Thus, if the digesta viscosity in the lumen is low, the diffusion rate in the lumen would be faster, as well as the absorption of molecules from the lumen. In our study, the ingestion of PDX increased the amount of SCFAs in the cecum and the cecal surface area, and we observed a faster digestion rate due to reduced cecal digesta viscosity, leading to an increased the SCFA absorption rate from the cecum into the venous blood, which ultimately leads to an increased salivary IgA flow rate.

### 4.1. Production of SCFAs in the Cecum by PDX Ingestion

In this study, PDX ingestion increased the rate of the SCFA absorption from the cecum and reduced the SCFA concentration in the cecal digesta (Table 1). It has already been reported that the intake of PDX reduces the SCFA concentration in cecal contents in vivo [20,36]. However, in vitro studies on large intestinal bacteria batch cultures have shown that the addition of PDX increases SCFA production [37,38]. Therefore, the measured SCFA concentration in the cecal digesta does not reflect the actual amount of SCFAs produced in the cecum that would be disappeared quickly.

### 4.2. SCFAs’ Role in Salivary IgA Activation

The Bayesian network analysis showed that the salivary IgA flow rate, normalized on the submandibular gland tissue weight, was directly affected by the SCFA concentration in portal vein blood (Figure 5). Reportedly, blood SCFA levels are sufficient to activate G-protein-coupled receptors (GPR) 41 and GPR43, to trigger autonomic nerve stimulation [39,40,41]. In addition, Carpenter et al. predicted that autonomic nerve stimulation would significantly elevate salivary IgA levels [42,43]. In the present study, both cecal digesta amount and portal vein concentration of total SCFAs, acetate, and n-butyrate were higher in the PDX group compared to the control group (Table 2 and Table 3). Therefore, GPR41 and GPR43 seemingly activate autonomic nerves, which is accompanied by an increased salivary IgA flow rate.

### 4.3. pIgR and Sympathetic and Parasympathetic Nerves

The transportation of IgA into the saliva is dependent upon the expression of epithelial receptor pIgR on salivary cells [42]. Sympathetic stimulation increases pIgR expression in rat submandibular gland [44] and rats subjected to preganglionic parasympathectomy showed a reduced pIgR membrane expression in the submandibular gland [45]. In addition, Wada et al. reported that sympathetic nerve stimulation would increase pIgR expression also in mouse submandibular glands [46]. Thus, the regulation of pIgR expression in the salivary gland is under the control of sympathetic and parasympathetic nerves. In this study, we observed that pIgR mRNA expression in the submandibular gland was higher in the PDX group than that in the control group (Figure 4A). Thus, our data suggest that pIgR upregulation is mediated by sympathetic and parasympathetic nerves, which are activated by the higher SCFA levels in the blood following PDX intake.

Indeed, the Bayesian network analysis shows that the pIgR mRNA expression in the submandibular gland is affected by the amount of SCFAs in the cecal digesta (Figure 5). Moreover, it has been reported that, in fasting rats, the injection of SCFAs into the large intestine stimulates epithelial cell proliferation through the activation of the sympathetic nervous system [47]. The results of the present study suggest a similar mechanism, in which the increased amount of SCFAs in the cecum, as well as increased SCFA concentration in the blood, stimulate sympathetic nerves, which in turn increase the expression of pIgR in the submandibular gland.

### 4.4. Differences Between Regulatory Mechanisms of pIgR Expression in Salivary Glands and pIgR Expression in the Intestinal Tract

Ingestion of indigestible carbohydrates has previously been reported to increase pIgR expression in the intestinal tract [18,22,48]. In the intestinal tract, increased expression of pIgR can be caused by the direct attachment of bacteria to the epithelial cell surface [49]. In particular, lipopolysaccharide (LPS) molecules on the bacterial wall of enterobacteria stimulate the release of various cytokines by the host innate and adaptive immune cells, such as IFN-γ, IL-1, IL-4, IL-17, tumor necrosis factor (TNF), and lymphotoxin β (LT-β) [49]. These stimuli activate the NF-kB pathway, which in turn increases intestinal pIgR expression [49]. Similarly, Genda et al. reported that FOS intake in rats caused an increase in IFN-γ, IL-17A, and TNFα mRNA expression, as well as pIgR mRNA expression, in the cecum [18].

We aimed to identify whether the same mechanism could apply for the regulation of pIgR expression in salivary glands. Salivary glands are exocrine glands that secrete saliva onto the mucosal surface of the oral cavity. Under normal conditions, oral bacteria cannot enter into salivary glands [50]. Moreover, salivary gland cells are intimately regulated by the autonomic nervous system [51]. Thus, it is unlikely that pIgR could be activated following cytokine release by immune cells. Indeed, in this study, we observed no difference between the two groups in the concentration of IFN-γ and IL-17A in the submandibular gland, although the submandibular gland pIgR mRNA levels were higher in the PDX group than in the control group (Figure 4). In addition, no correlation was observed between the salivary IgA flow rate and the expression of IFN-γ and IL-17A (Table 6). Thus, our data suggest that, differently from the intestinal pIgR regulation, in the salivary gland the increase of pIgR expression following the ingestion of indigestible carbohydrates such as PDX is under the control of the autonomic nerve stimulation.

In this study, we only measured SCFA levels in the portal vein, to determine the SCFA absorption from the cecum. However, we did not collect the circulating blood and thus we could not measure the SCFA levels in the circulating blood. However, it has been previously reported that circulating SCFA levels in mice are increased by the ingestion of a high fiber diet [39] and that the SCFA concentration in the portal vein is positively correlated with SCFA levels in the circulating blood [52]. Hence, the measurement of SCFA levels in the portal vein can be used instead of the levels in the circulating blood.

In this experiment, we investigated the effect of short chain fatty acids absorbed in the cecum on the rate of IgA secretion in whole saliva. In experiments in which rodent whole saliva is collected, it is common to collect the whole saliva stimulated with pilocarpine to examine the submandibular gland, which can be examined for both serous cell and mucous cell effects [8,53]. However, unstimulated saliva and changes in the parotid gland were not examined. Further investigation is warranted.

The Bayesian network analysis of causal effects showed that pIgR expression levels in the submandibular gland had no direct effect on the salivary IgA flow rate (Figure 5). This is in apparent contrast with previous studies showing that the IgA produced in salivary glands is transported into the saliva via pIgR [45], as well as the positive correlation between pIgR expression in salivary glands and salivary IgA flow rate [8,22], which was also confirmed in our study (Table 4). This direct effect could be hidden in the Bayesian network by the influence of parameters with a stronger correlation with salivary IgA flow rate, such as the concentration of SCFAs in the portal vein, cecal digesta water content, and cecal tissue weight (Table 4).

## 5. Conclusions

In conclusion, we found that PDX ingestion increased the salivary IgA flow rate and pIgR expression levels in the salivary gland. The increase in salivary IgA secretion was directly attributable to the higher levels of SCFAs rapidly absorbed from the cecum, rather than the concentration of SCFAs in the cecal digesta. The rapid absorption of SCFAs from the cecum was affected by an increase in the cecal tissue weight, cecal digesta weight, amount of SCFAs in cecal digesta, cecal digesta water content, and a decrease in cecal digesta viscosity caused by PDX-stimulated fermentation. Moreover, our results suggest that, unlike in the intestinal tract, the increase in pIgR expression in salivary glands following PDX ingestion is not affected by increased cytokines.

## Figures and Tables

**Figure 1 nutrients-12-01745-f001:**
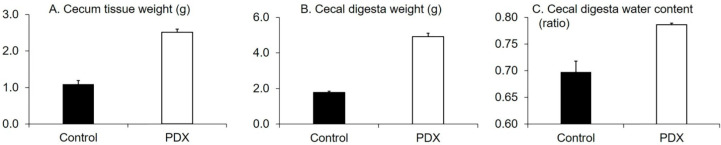
Effects of polydextrose (PDX) addition on the cecum tissue weight (**A**), cecal digesta weight (**B**), and cecal digesta water content (**C**). n = 6 per group. Data are expressed as means (histogram bars) and SEM (error bars). Cecum tissue weight, cecal digesta weight, and cecal digesta water content in the PDX group were higher than those in the control group (*p* < 0.0001, 0.0001, and *p* = 0.007, respectively, Welch’s *t*-test).

**Figure 2 nutrients-12-01745-f002:**
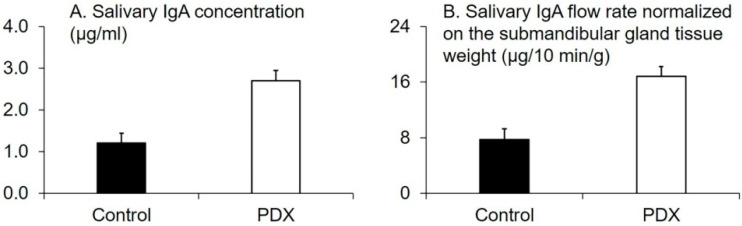
Effects of polydextrose (PDX) addition on the salivary immunoglobulin A (IgA) concentration (**A**) and salivary IgA flow rate normalized on the submandibular gland tissue weight (**B**). n = 6 per group. Data are expressed as means (histogram bars) and SEM (error bars). Salivary IgA concentration and flow rate normalized on the submandibular gland tissue weight in the PDX group were higher than those in the control group (*p* = 0.001 and 0.001, respectively, Welch’s *t*-test).

**Figure 3 nutrients-12-01745-f003:**
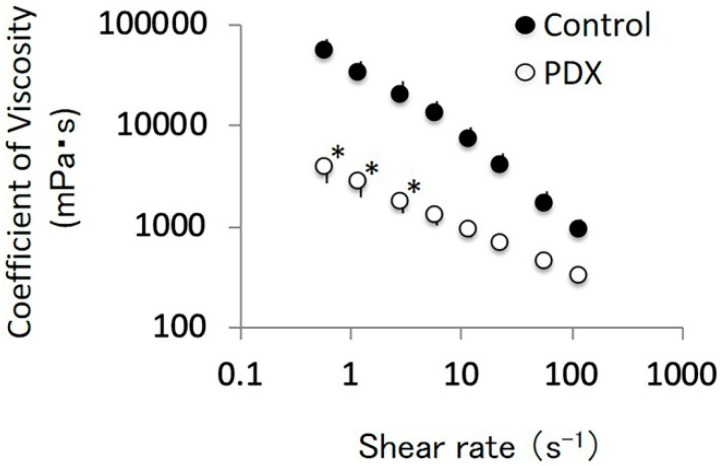
Effects of polydextrose (PDX) addition and shear rate on the cecal digesta viscosity. n = 6 per group for each shear rate. Data are expressed as means (histogram bars) and SEM (error bars). We found a statistically significant interaction between PDX addition and shear rate for the cecal digesta viscosity (*p* = 0.001, two-way ANOVA). * *p* < 0.05 versus control group at each shear rate using the Tukey–Kramer method.

**Figure 4 nutrients-12-01745-f004:**
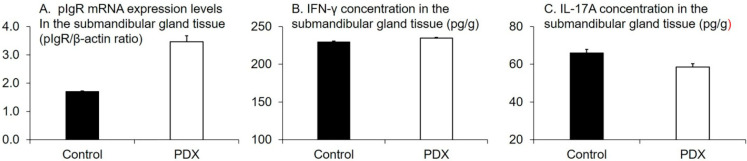
Effects of polydextrose (PDX) addition on the polymeric immunoglobulin receptor (pIgR) mRNA expression levels in the submandibular gland tissue (**A**), the interferon γ (IFN-γ) concentration in the submandibular gland tissue (**B**), and the interleukin 17A (IL-17A) concentration in the submandibular gland tissue (**C**). n = 6 per group. Data are expressed as means (histogram bars) and SEM (error bars). pIgR mRNA expression levels in the submandibular gland tissue in the PDX group was higher than that in the control group (*p* = 0.02, Welch’s *t*-test). We found no significant differences in IFN-γ and IL-17A concentration in the submandibular gland tissue between the two groups (*p* = 0.8 and 0.3, respectively, Welch’s *t*-test).

**Figure 5 nutrients-12-01745-f005:**
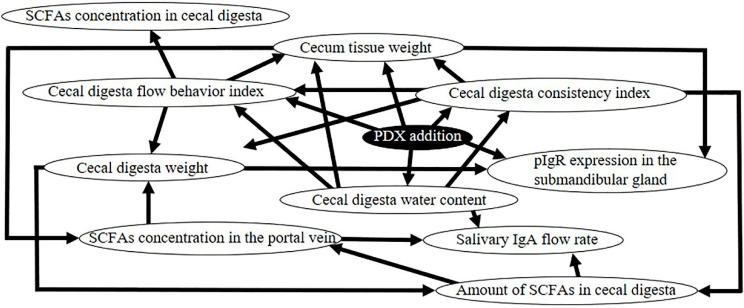
Bayesian network showing the causal effects between the following parameters, upon PDX ingestion: Salivary IgA flow rate, SCFA concentration in the portal vein, SCFA concentration in the cecal digesta, amount of SCFAs in cecal digesta, cecum tissue weight, cecal digesta weight, cecal digesta water content, cecal digesta consistency index, cecal digesta flow behavior index, and pIgR expression in submandibular gland. Causes and effects are indicated by arrowheads and lines, respectively.

**Table 1 nutrients-12-01745-t001:** Short-chain fatty acids (SCFAs) concentration in cecal digesta, 4 weeks after feeding (mmol/kg digesta).

Acids	Control		PDX *		*p*-Value ^†^
	Mean	SEM	Mean	SEM	
Acetate	23.3	1.19	15.7	1.59	0.004
Propionate	7.43	0.58	3.81	0.56	0.001
*n*-Butyrate	5.36	1.05	5.14	0.68	0.9
SCFAs total	36.1	1.97	24.7	2.75	0.008

n = 6; * PDX: polydextrose; ^†^ calculated using Welch’s *t*-test.

**Table 2 nutrients-12-01745-t002:** Short-chain fatty acids (SCFAs) concentration in the portal vein, 4 weeks after feeding (μmol/L).

Acids	Control		PDX *		*p*-Value ^†^
	Mean	SEM	Mean	SEM	
Acetate	227.2	16.48	305.8	13.84	0.005
Propionate	43.62	3.270	42.80	6.722	0.9
*n*-Butyrate	27.93	3.318	46.90	6.221	0.03
SCFAs total	298.7	19.20	395.5	16.92	0.004

n = 6; * PDX: polydextrose; ^†^ calculated using Welch’s *t*-test.

**Table 3 nutrients-12-01745-t003:** Amounts of short-chain fatty acids (SCFAs) in cecal digesta, 4 weeks after feeding (μmol).

Acids	Control		PDX *		*p*-Value ^†^
	Mean	SEM	Mean	SEM	
Acetate	41.2	2.78	77.8	9.42	0.005
Propionate	13.1	1.24	18.9	3.08	0.06
*n*-Butyrate	9.63	2.08	25.4	3.73	0.003
SCFAs total	63.9	5.04	122	15.9	0.007

n = 6; * PDX: polydextrose; ^†^ calculated using Welch’s *t*-test.

**Table 4 nutrients-12-01745-t004:** Correlation between salivary IgA flow rate normalized on the submandibular gland tissue weight and other experimental parameters used in the study.

	Salivary IgA Flow Rate Normalized on the Submandibular Gland Tissue Weight		
	r_s_ *	*p*-Value	n
Concentration of SCFAs in portal vein blood	0.88	0.0002	12
Weight of cecal tissue	0.76	0.004	12
Water content of cecal digesta	0.76	0.005	12
pIgR expression level in submandibular gland	0.66	0.02	12
Weight of cecal digesta	0.60	0.04	12
Concentration of SCFAs in cecal digesta	−0.53	0.09	12

SCFAs: Short-chain fatty acids, which was the sum of the concentration of acetate, propionate, and *n*-butyrate. * Spearman’s rank correlation coefficient.

**Table 5 nutrients-12-01745-t005:** Correlation between the short-chain fatty acid (SCFA) concentration in the portal vein and the cecal tissue weight, cecal digesta weight, amount of SCFAs in cecal digesta, and cecal digesta water content.

	Concentration of SCFAs in the Portal Vein		
	r_s_ *	*p*-Value	n
Cecal tissue weight	0.74	0.006	12
Cecal digesta weight	0.69	0.01	12
Amount of SCFAs in cecal digesta	0.66	0.02	12
Cecal digesta water content	0.63	0.03	12

SCFAs: Short-chain fatty acids, which was the sum of the concentration of acetate, propionate, and *n*-butyrate. * Spearman’s rank correlation coefficient.

**Table 6 nutrients-12-01745-t006:** Correlation between the pIgR expression level in the submandibular gland tissue and the cecal tissue weight, cecal digesta weight, short-chain fatty acid (SCFA) concentration in the portal vein, and concentration of interferon-γ (IFN-γ) and interleukin-17A (IL-17A) in the submandibular gland tissue.

	pIgR Expression Level in Submandibular Gland Tissue		
	r_s_ *	*p*-Value	n
Cecal tissue weight	0.87	0.0003	12
Cecal digesta weight	0.77	0.003	12
Concentration of SCFAs in the portal vein	0.71	0.009	12
Concentration of IFN-γ in the submandibular gland tissue	−0.046	0.9	12
Concentration of IL-17A in the submandibular gland tissue	−0.028	0.9	12

SCFAs: Short-chain fatty acids, which was the sum of the concentration of acetate, propionate, and *n*-butyrate. * Spearman’s rank correlation coefficient.

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
