# Peer review of "Faster Short-Chain Fatty Acid Absorption from the Cecum Following Polydextrose Ingestion Increases the Salivary Immunoglobulin A Flow Rate in Rats"

_nutrients, 2020, doi:10.3390/nu12061745_

Round 1
Reviewer 1 Report
The manuscript describes effect of polydextrose ingestion on salivary IgA production through pIgr. It appears to be an extension of a previous work by the authors where FOS/ Pdx+lactitol were used and does not add substantially to the field of dietary fiber benefits. The concept of applying Bayesian network analysis is attractive.
There are several major concerns besides language errors. Composition of diet does not add upto 1000 in Pdx group. Why was corn oil reduced to 40g/kg? How the food intake was measured is not described. As the manuscript is mainly based on salivary IgA, a description of method of measurement of flow rate is crucial. In discussion section several suggestions are included without any basis. Like role of parasympathetic and sympathetic nerves in IgA production and activation of GPR43 and 41.
Author Response
We thank for appropriate advice from you very much. The corrected parts are marked with yellow lines in the manuscript.
We made a mistake in the description of the diet composition of the PDX group. The correct content of corn oil was 50.0 g/ kg. We have corrected the numbers in ‘Table S1.’ .
According to Reviewer 1’s suggestion, the method for measuring the feed intake was added. ‘The feed intake of rats was determined by measuring the weight of the residual feed every three days.’ (Line 95-96, Page 2).
According to Reviewer 1’s suggestion, the method of measuring salivary IgA flow rate is added. ‘The whole saliva flowed out by pilocarpine (8 mg/kg body weight) injection was collected with a micropipette for 10 min.’ (Line 121-122, Page 3). ‘Saliva weight (g) collected for 10 min was measured, and the specific gravity of saliva was assumed to be 1.00 g/ml, and the salivary flow rate (ml/10min) was calculated. Salivary IgA flow rate (μg/10min) was calculated by multiplying salivary IgA concentration (μg/ml) by salivary flow rate (ml/10min)’ (Line 124-128, Page 3).
According to Reviewer 1’s suggestion, hypotheses that are not based on the basis of the ‘Discussion’ have been largely deleted. ‘4.2. SCFAs role in salivary IgA activation The Bayesian network analysis showed that the salivary IgA flow rate, normalized on the submandibular gland tissue weight, was directly affected by the SCFA concentration in portal vein blood (Figure 5). Reportedly, blood SCFA levels are sufficient to activate G-protein-coupled receptors (GPR) 41 and GPR43, to trigger autonomic nerve stimulation [39-41]. In addition, Carpenter et al. predicted that autonomic nerve stimulation would significantly elevate salivary IgA levels [42, 43]. In the present study, both cecal digesta amount and portal vein concentration of total SCFAs, acetate, and n-butyrate were higher in the PDX group compared to the control group (Table 2, 3). Therefore, GPR41 and GPR43 seemingly activate autonomic nerves, which is accompanied by an increased salivary IgA flow rate.’ (Line 373-382. Page 11).
Reviewer 2 Report
The authors provided solid evidence that the role of short-chain fatty acid for the enhancement of salivary IgA antibody responses. Thus, the present study is important to show how nutrition affects the oral immune system and maintain one's oral health.
Minor Points:
Although the authors introduced that decreased salivary IgA antibody level increases the risk of upper respiratory infection, the current study did not directly prove this concept. Thus, it would be better to remove the last sentence in the Abstract section (ultimately contributing to URTI prevention).
Line 385: misspelled
Author Response
We thank for appropriate advice from you very much. The corrected parts are marked with yellow lines in the manuscript.
According to Reviewer 2’s suggestion, we have corrected the last sentence of the abstract. ‘which is accompanied by an increase in SCFA levels in the blood, and ultimately leads to increased salivary IgA levels.’ (Line 35-36, Page 1).
The Line 385 word in the original manuscript was misspelled. However, the sentence was deleted by another reviewer's suggestion.
Reviewer 3 Report
The article is interesting from a clinical and scientific point of view. I have a few questions for the authors asking for some ambiguities.
First, please provide a detailed description of saliva collection.
Why only submandibular glands were collected, not parotid. From what I understand, the authors took stimulated saliva, which they then used for determinations. The submandibular salivary glands produce almost and exclusively unstimulated saliva, and the parotid gland -stimulated saliva. Therefore, I do not understand why the submandibular gland and stimulated saliva were taken.
When providing p values, round off the values to a maximum of two decimal places.
There are too many hypotheses in the discussion that are not supported by the results of the authors' research.
Author Response
We thank for appropriate advice from you very much. The corrected parts are marked with yellow lines in the manuscript.
According to Reviewer 3’s suggestion, we have added additional details on saliva collection methods. ‘The whole saliva flowed out by pilocarpine (8 mg/kg body weight) injection was collected with a micropipette for 10 min.’ (Line 121-122, Page 3).
According to Reviewer 3’s suggestion, we added the reason why we sampled only whole saliva and submandibular glands and not the parotid glands. ‘In this experiment, we investigated the effect of short chain fatty acids absorbed in the cecum on the rate of IgA secretion in whole saliva. In experiments in which rodent whole saliva is collected, it is common to collect the whole saliva stimulated with pilocarpine to examine the submandibular gland, which can be examined for both serous cell and mucous cell effects [8, 53]. However, unstimulated saliva and changes in the parotid gland were not examined. Further investigation is warranted.’ (Line 431-436, Page 12).
According to Reviewer 3’s suggestion, we changed the number of digits in all p-values.
According to Reviewer 3’s suggestion, we have significantly removed non-result-based hypotheses from the discussion. ‘4.2. SCFAs role in salivary IgA activation The Bayesian network analysis showed that the salivary IgA flow rate, normalized on the submandibular gland tissue weight, was directly affected by the SCFA concentration in portal vein blood (Figure 5). Reportedly, blood SCFA levels are sufficient to activate G-protein-coupled receptors (GPR) 41 and GPR43, to trigger autonomic nerve stimulation [39-41]. In addition, Carpenter et al. predicted that autonomic nerve stimulation would significantly elevate salivary IgA levels [42, 43]. In the present study, both cecal digesta amount and portal vein concentration of total SCFAs, acetate, and n-butyrate were higher in the PDX group compared to the control group (Table 2, 3). Therefore, GPR41 and GPR43 seemingly activate autonomic nerves, which is accompanied by an increased salivary IgA flow rate.’ (Line 373-382. Page 11).
Round 2
Reviewer 1 Report
No further revision is required.
Reviewer 3 Report
All my comments have been taken into account. That is why I believe that the article can be published in its current form